# Optimizing Blood Glucose Control through the Timing of Exercise in Pregnant Individuals Diagnosed with Gestational Diabetes Mellitus

**DOI:** 10.3390/ijerph20085500

**Published:** 2023-04-13

**Authors:** Áine Brislane, Ly-Anh Reid, Gyan Bains, Kelly Greenwall, Rshmi Khurana, Margie H. Davenport

**Affiliations:** 1Program for Pregnancy & Postpartum Health, Neurovascular Health Lab, Faculty of Kinesiology, Sport, and Recreation, Women and Children’s Health Research Institute, Alberta Diabetes Institute, University of Alberta, Edmonton, AB T6G 2E1, Canada; 2Departments of Medicine and Obstetrics & Gynaecology, Faculty of Medicine, Women and Children’s Health Research Institute, University of Alberta, Edmonton, AB T6G 2E1, Canada

**Keywords:** pregnancy, gestational diabetes, walking, postprandial, glucose

## Abstract

This study aimed to evaluate the effectiveness of moderate intensity walking on postprandial blood glucose control for pregnant individuals with (GDM) and without gestational diabetes mellitus (NON-GDM). Using a randomized cross-over design, individuals completed 5 days of exercise (three 10-min walks immediately after eating (SHORT), or one 30-min walk (LONG) outside of 1 h after eating). These protocols were preceded and separated by 2-days of habitual exercise (NORMAL). Individuals were instrumented with a continuous glucose monitor, a physical activity monitor for 14-days, and a heart rate monitor during exercise. Participants completed a physical activity enjoyment scale (PACES) to indicate their protocol preference. The GDM group had higher fasting, 24-h mean, and daily peak glucose values compared to NON-GDMs across all conditions (effect of group, *p* = 0.02; *p* = 0.02; *p* = 0.03, respectively). Fasting, 24-h mean, and daily peak glucose were not influenced by the SHORT or LONG exercise (effect of intervention, *p* > 0.05). Blood glucose values were higher among the GDM group for at least 1 h after eating, yet the exercise intervention had no effect on 1 or 2 h postprandial glucose values (effect of intervention, *p* > 0.05). Physical activity outcomes (wear time, total activity time, and time spent on each intensity) were not different between the groups nor interventions (effect of group, *p* > 0.05; effect of intervention, *p* > 0.05,). There were no differences between the groups or interventions for the PACES score (effect of group, *p* > 0.05; effect of intervention, *p* > 0.05). To conclude, there were no differences between the groups or exercise protocols on blood glucose control. More research is warranted to elucidate higher exercise volumes in this outcome for individuals with GDM.

## 1. Introduction

Gestational Diabetes Mellitus (GDM) affects 3 to 20% of Canadian pregnancies [1] and is defined as glucose intolerance with the onset or first recognition in pregnancy [2]. GDM occurs when the normal pregnancy-related physiological insulin resistance becomes imbalanced, resulting in maternal blood glucose levels that rise to pathological values [2]. During a healthy pregnancy, insulin resistance increases with advancing gestation [3]. However, this effect is pronounced in women with GDM, who have markedly higher insulin resistance compared to their healthy counterparts and, consequentially, elevated blood glucose [3]. This condition can contribute to immediate maternal and fetal complications including pre-eclampsia, caesarian delivery, birth weight >90th percentile, and neonatal hypoglycemia [4,5,6,7].

In addition to the immediate health risks, GDM has long-term consequences for maternal and fetal health compared to healthy counterparts. For example, women with a history of GDM have been shown to be at a higher risk of obesity, hypertension, and dyslipidemia, with respective risk ratios of 2.4, 7.5, and 2.4 [8]. A hazard ratio for cardiovascular disease events following a history of GDM has also been reported to be 1.71 [9]. Elsewhere, adult offspring born to women with diet-treated GDM were twice as likely to be overweight compared to an unexposed reference group [10]. The authors also demonstrated a fourfold increased risk of metabolic syndrome, a cluster of conditions including abdominal obesity, abnormal cholesterol levels, abnormal triglyceride levels, high blood pressure, and/or high blood sugar. Given the health risks posed to both the mother and baby with a diagnosis of GDM, interventions are necessary during this critical time to reduce the risk of cardiovascular disease, progression to type 2 diabetes, and adverse neonatal outcomes [11].

Front-line therapies in GDM involve daily blood glucose monitoring, dietary modifications regarding carbohydrate intake, pharmacological treatment (most commonly insulin or Metformin), and recommendations for aerobic exercise. Specifically, pregnant individuals with GDM are advised to complete 150 min of moderate-intensity exercise per week [12]. Recently, the American College of Obstetrics and Gynecology (ACOG) suggested that a 10–15 min walk after each meal may improve glycemic control, an outcome that is of particular importance for individuals with GDM [2]. However, this exercise recommendation is largely based on expert opinion and not empirical evidence. As such, there is a need to distinguish between the effectiveness of intermittent and accumulated exercise bouts that equate to 150 min per week regarding glycemic control for individuals with GDM.

Previous research has demonstrated that the control of postprandial blood glucose values may be central to improving maternal and fetal outcomes in GDM. A study from de Veciana et al. (1995) showed that individuals who self-administered insulin therapy based on postprandial and not pre-prandial blood glucose values delivered babies of lower birth weights, had a lower incidence of neonatal hypoglycemia, and were less likely to have a caesarean section [13]. As such, there is some rationale for exploring the timing of alternative therapies postprandial.

Lifestyle interventions that aim to improve blood glucose control in GDM have largely comprised changes to the diet with and without exercise. Allehdan et al. (2019) collated data from eight trials that compared the effect of diet plus exercise to the effect of diet-only interventions on postprandial blood glucose [14,15,16,17,18,19,20,21,22] Among the trials, six evidenced lower postprandial blood glucose control when diet and exercise interventions (three aerobic, one yoga, one resistance and one aerobic and resistance exercise combined) were combined versus dietary changes only [16,17,19,20,21,22]. This implies that the exercise stimulus is a primary driver, via non-insulin-mediated mechanisms, in helping to manage postprandial blood glucose in GDM. It is understood that exercise increases the rate of glucose uptake into the skeletal muscle during and following exercise. This increased uptake occurs due to the translocation of the glucose transport protein GLUT-4 from intracellular sites to the sarcolemma and T-tubules. This action increases the sites at which glucose can diffuse into the muscle cell and thus reduces the level of glucose in the blood [23,24]. To date, the isolated effect of exercise on blood glucose control in GDM is poorly characterized.

Christie et al. (2022) investigated, for the first time, the effect of three 10-min walks throughout the day after each meal, with 30 min of continuous walking at any time of the day [25]. This study observed no influence of either walking strategy on blood glucose control in GDM; however, it is unknown if this response would be comparable to that of individuals without GDM. To build on this knowledge, we herein perform a similar protocol with the addition of a normoglycemic control group to elucidate any group differences that may emerge. The aim of this study was to evaluate the effectiveness of postprandial versus daily exercise (outside of the 1 h post-meal window) regarding acute and 24 h blood glucose control for pregnant individuals with and without gestational diabetes mellitus.

## 2. Methodology

### 2.1. Study Design

This study was a randomized, cross-over study design that compared two different exercise prescriptions for pregnant individuals with and without GDM. The participants were randomized to start with one of two study conditions using a randomization scheme (www.sealedenvelope.com, accessed on 7 March 2023) that was followed by the complementary condition. The randomization was completed by a researcher external to the investigative team, and assignments were sequentially provided when each participant consented. This study was conducted in accordance with the Declaration of Helsinki, was approved by the Ethics Committee at the University of Alberta (University of Alberta ethics protocol Pro00097525), and is registered as a clinical trial at ClinicalTrials.gov, accessed on 7 March 2023 (Registration: NCT05256615).

### 2.2. Participants

To be included in the study, the participants were required to be residents of Canada, be pregnant with one baby, and have either a diagnosis of GDM or an uncomplicated pregnancy. The participants were recruited after 20 weeks’ gestation since GDM is generally screened for and diagnosed toward the end of the second trimester of pregnancy. Individuals were excluded if they had absolute contraindications to prenatal exercise, as identified by the PARmed-X for Pregnancy. These contraindications included premature labor, placenta previa, pregnancy-induced hypertension, pre-eclampsia, and high-order pregnancy uncontrolled systemic disease including cardiovascular and respiratory disorders [12,26]. The participants were recruited by a physician working in a GDM clinic and via social media advertisements on Facebook and Instagram. Posters and pre-recorded video presentations were also created and distributed to relevant provincial and national health and education clinics specializing in diabetes and pregnancy.

### 2.3. Study Structure

Individuals that expressed interest in the study were provided with a detailed information sheet and were invited to speak with a researcher about the study. In this initial conversation, the study was explained, and all queries (if any) were addressed. For those interested in participating, written consent was obtained, and participants were assigned a study identification number. The participants completed a Health History Questionnaire before receiving a package in the mail containing all study materials (i.e., heart rate monitor, accelerometer, continuous glucose monitor, and study booklet). Once a study pack had been received, a video call was arranged between the researcher and participant via an online and secure platform (https://doxy.me/en/, accessed on 7 March 2023). The purpose of this call was for the researcher to guide the participant through each component of the pack and to ensure competency regarding the wearables and the walking protocol.

The study period began with two days of normal daily physical activity (NORMAL), followed by five consecutive days of the first intervention condition and then a washout period of, again, two days of NORMAL daily physical activity, followed by five days of the second intervention condition (Figure 1). The interventions comprised three 10 min walks per day for five days (SHORT) or one 30 min walk each day for 5 days (LONG).

### 2.4. Exercise Protocol

Following the completion of two NORMAL days to establish baseline blood glucose and activity monitoring, all participants completed the SHORT and LONG protocol in a randomized order. This randomized approach was taken to ensure that the interpretation of the results was not limited by the sequence of the protocol. Women randomized to the SHORT condition were asked to complete a 10 min walk within the first hour after breakfast, lunch, and dinner, totaling three 10 min walks per day for five days. Those randomized to the LONG condition were asked to complete 30 min of walking at any time of day, other than the hour immediately following breakfast, lunch, or dinner, for five days. Between interventions, the participants completed two more days of NORMAL activity that fulfilled a washout period between protocols. Following the two-day washout period, the participants were asked to complete the complementary exercise protocol for five days.

Throughout the study, the participants wore a flash glucose monitor to measure interstitial fluid glucose and an accelerometer to measure physical activity. Compliance with the exercise intervention was quantified by totaling the number of sessions completed by each participant and expressed as a percentage of the prescribed number of sessions. Exercise adherence was assessed by confirming the heart rate recorded for each session. An exercise session was adhered to if the heart rate was within the prescribed range of 121–146 beats per minute (bpm).

Walking was the prescribed modality because of its accessibility, feasibility, and low cost [26]. In both exercise conditions, the participants were asked to walk at a self-selected light-to-moderate physical activity intensity and wear a heart rate monitor (Polar) in order to confirm they were in the prescribed intensity range (moderate; 121–146 bpm) [12]. Both exercise intervention conditions met the current physical activity recommendations for pregnant women of 150 min/week of moderate-intensity aerobic exercise [12].

### 2.5. Instrumentation

#### 2.5.1. Flash Glucose Monitor

The participants were provided with a Flash Glucose Monitoring System (Abbott, Chicago, IL, USA) to wear for the entirety of the study period—a device that has previously been evaluated in pregnant women with type 1, type 2, and gestational diabetes. [27]. This minimally invasive, small monitoring device adheres to the skin at the triceps, and its application was coached by the researcher during video calls. It detects and records interstitial fluid glucose levels at 5 min intervals for up to two weeks. The data are recorded internally and can be viewed on a graph on the accompanying handheld reader once the senor is scanned. The continuous glucose monitor (CGM) was applied to the back of the upper arm once cleaned with an alcohol wipe, with assistance from the researcher via video call. The device comes with an applicator containing a needle, which is joined with the sensor to facilitate application. Once the arm was dry, the sensor applicator was placed over the site, and a firm push guided the needle and filament under the skin. The needle was automatically and immediately removed. One hour after application, the participants followed written instructions to activate the sensor on a handheld reader. The participants were able to scan their CGM and view their glucose graph throughout the duration of the study. At the end of the study period, the participants returned the sensor and reader. Participants with GDM were asked to follow their healthcare provider’s guidance for GDM management, including continuing their capillary blood glucose measurements, diet recommendations, and medications as prescribed.

Data from the CGM devices were downloaded to Microsoft Excel files using the FreeStyle Libre Software Version 1.0 software (Abbott, Chicago, IL, USA) and analyzed offline. Interstitial fluid glucose was measured and used as a proxy for blood glucose levels. The primary outcomes were 1 and 2 h postprandial glucose values after the start of each meal (breakfast, lunch, dinner). Secondary outcomes included fasting (value upon awakening), mean 24 h (midnight to midnight), peak and nadir glucose, time in target (3.3–7.8 mmol/L), time spent < 3.3 mmol/L, and time spent > 7.8 mmol/L. Daily mean 1 and 2 h postprandial outcomes were calculated using days 2 to 5 of the SHORT condition and days 2 to 5 of the LONG condition. Daily mean 24 h, peak and nadir glucose, time in target, time < 3.3 mmol/L, and >7.8 mmol/L were calculated using days 2 to 5 of the first condition completed and days 2 to 4 of the second condition completed, as the glucose monitor stopped recording partway through the 14th day of the study period. The first day of each condition was excluded in order to not use data collected prior to the exercise stimulus. While the CGM records interstitial fluid glucose values in five-minute intervals, only 15 min average blocks are able to be exported. The area under the curve was also calculated within 1 and 2 h of each meal to account for glucose control. The participants’ values for each outcome within each condition (NORMAL, SHORT, LONG) were averaged and contributed to the groups’ means.

#### 2.5.2. Food Record Intake

In order to determine the caloric and nutritional intake as well as the time of meals, the participants were asked to keep a log of their food consumption for the entire study period (14 days). The participants recorded when they ate their meals and the specifics of the foods they consumed such as the type, brand, amount, and condiments and any ingredients used for cooking (e.g., butter, oil). They were instructed to be as detailed as possible, describing individual ingredients for all of their food intake. The participants were also provided with a guide to determine the amount (volume) of food items or ingredients they consumed, rather than weighing them. The food intake records were returned to the laboratory at the end of the study period, and information regarding diet intake was derived from these records.

Dietary intake was derived from the participant’s food intake records and entered into the Food Processor Program (ESHA Research, Salem, OR, USA). Variables of interest across conditions and between groups included the mean daily caloric intake, protein, fat, and carbohydrate intake. Values for each outcome within each condition were averaged per participant and included four days for the NORMAL condition and five days for each of the SHORT and LONG conditions.

### 2.6. Accelerometer

The participants were asked to wear an accelerometer (Actigraph wGT3X-BT Monitor, Actigraph LLC, Pensacola, FL, USA) for the entire study period (14 consecutive days and nights) to record 24 h physical activity measurements. This information was collected to determine the overall physical activity and movement behaviors (including activity intensity). The participants wore the accelerometer on their waist during the day and on a wrist strap at night. Data were downloaded onto specific software (ActiLife 6, Actigraph LLC) and analyzed for activity levels.

The data collected were used to evaluate the durations (summed durations of accelerations) and intensity (magnitude of accelerations) of their physical activity and the caloric expenditure throughout the waking wear time [28]. To determine intensity, Freedson accelerometer count ranges were used: sedentary (<100 counts per minute (cpm)), light activity (100–1951 cpm), and moderate to vigorous physical activity (≥1952 cpm) (Freedson et al., 1998). Non-wear times were confirmed using activity logs. Variables of interest included the active energy expenditure (kcals), average wear time per day, time spent sedentary, time spent in light-, moderate-, vigorous-, and very vigorous-intensity activity, and total time spent in physical activity. The values for each outcome within each condition were averaged and contributed to each group’s means. Only days with >600 min of wear time were included.

### 2.7. Heart Rate Monitor

The participants wore a heart rate monitor chest strap (Polar, Kempele, Finland) during their walking sessions to confirm they were in the prescribed intensity range (121–146 bpm). The chest strap is worn around the chest, with the sensor placed just inferior to the sternum. It wirelessly connects to the Polar Beat mobile application (downloaded through the Apple App Store or Google Play), which displays the heart rate in beats per minute (bpm). This device was returned to the researchers at the end of the study period. Due to technical barriers, the heart rate was only documented via Polar Flow for 17 participants in total (GDM, *N* = 9; NON-GDM, *N* = 8). Therefore, exercise compliance and adherence could only be evaluated for these individuals.

### 2.8. Questionnaires

The participants were asked to complete a Health History Questionnaire (HHQ) in order to screen for any current absolute contraindications to exercise during pregnancy that might be considered unsafe or render a participant ineligible for the study. The HHQ also served to obtain participants’ demographic, anthropometric, and health information such as weight, height, ethnicity, parity, and other health concerns. The participants completed the questionnaire in an online format via Redcap.

The participants were also asked to complete a Physical Activity Enjoyment Scale (PACES), which assesses the extent to which individuals enjoy or dislike participating in any given physical activity. This scale has previously been assessed and is a reliable and valid tool for the assessment of the enjoyment of physical activity [29,30]. This questionnaire asks participants to select a ranking on a scale of one to seven between two opposing statements such as “I enjoy it” and “I hate it”. There are 18 total rankings to be completed. The rankings are added up to give the participants’ final scores out of a maximum of 126. A lower score would suggest low enjoyment of the given activity, and a higher score would suggest more enjoyment. The participants were asked to complete the questionnaire twice—once at the end of day 7 and once at the end of day 14—to compare the enjoyment of each condition.

### 2.9. Statistical Analysis

Avery and Walker (2001) found that low-intensity postprandial exercise resulted in a mean difference (±SD) in blood glucose of 0.3 ± 0.3 at 30 min post-exercise [31]. Based on these findings, we estimated that 12 women are required per group to observe a significant difference in postprandial blood glucose and increased the required sample size by 20% to account for study withdrawal (*N* = 15 per group; 80% power, α = 0.05; G*Power version 3.1.9).

Descriptive statistics were calculated, and an unpaired *t*-test was used to compare demographic data between the GDM and NON-GDM groups of all participants that enrolled in the study. An independent t-test was also used to evaluate exercise adherence within the LONG and SHORT protocols separately between the GDM and NON-GDM groups. PACES scores were analyzed in an identical fashion. The outcomes including the glucose, physical activity, and dietary intake from each condition (NORMAL, SHORT, and LONG) and group (GDM and NON-GDM) were compared using a general linear model (group by condition) with a post hoc Holm-Sidak test (Jamovi, Version 1.6.23.0, Sydney, Australia). The participants that did not progress to the second prescribed exercise intervention because of pregnancy complications were excluded from the analysis (PA, HR, dietary intake and CGM data). Outcomes within each group were compared between conditions, and outcomes between groups were also compared within each condition. Significance was accepted at *p* < 0.05.

## 3. Results

The participant characteristics are displayed in Table 1, which includes 17 enrolled in the GDM group and 16 enrolled in the NON-GDM group. Overall, the participants in the GDM and NON-GDM groups were similar in terms of age, anthropometrics, parity, and gestational age at the time of study participation. Three participants in the GDM group had a previous history of the condition. In the GDM group, eight were prescribed insulin (one of which was nocturnal only), one individual was taking metformin, six were prescribed dietary changes without exercise, and two were managing their diagnosis with diet with exercise. No participants reported adverse events in relation to physical activity, the CGM, or blood glucose levels. For subsequent analysis, three participants were excluded within the GDM group (pregnancy complications following consent (*N =* 1), prescribed bed rest within two days of starting the study (*N* = 1), and early labor, preventing progression to the second intervention (*N* = 1)). One individual was excluded from the NON-GDM group because food and physical activity diaries were not returned, and, therefore, CGM data could not be accurately analyzed.

### 3.1. Glucose Outcomes

#### 3.1.1. Daily Glucose Values (Fasting, 24 h, Peak, Nadir, Time in Target, Time < 3.3, Time > 7.8)

As expected, the GDM group exhibited higher fasting, 24 h mean, and daily peak glucose values (effect of group, *p* = 0.02; *p* = 0.02; *p* = 0.03 respectively) compared to the NON-GDM group across all conditions. Within groups, fasting, 24 h mean, and daily peak glucose were not influenced by the SHORT or LONG exercise (effect of intervention, *p* = 0.90, *p* = 0.69, and *p* = 0.99, respectively). Daily nadir glucose was similar between the GDM and NON-GDM groups (effect of group, *p* = 0.11) and was not influenced by the SHORT or LONG exercise (effect of intervention, *p* = 0.48, Table 2).

The time spent in hypo- (<3.3 mmol/L) and hyperglycemia (>7.8 mmol/L) was similar between the GDM and NON-GDM groups (effect of group, *p* = 0.98 and *p* = 0.17, respectively). Within the groups, the time spent at <3.3 mmol/L and >7.8 mmol/L was not affected by the SHORT or LONG exercise (effect of intervention, *p* = 0.73 and *p* = 0.94, respectively). Equally, the time spent at target glucose levels (3.3 to 7.8 mmol/L) was similar between the GDM and NON-GDM groups (effect of group, *p* = 0.60) and was not affected by the SHORT or LONG exercise (effect of intervention, *p* = 0.95, Table 2).

#### 3.1.2. Post-Breakfast Glucose Values

The absolute postprandial blood glucose within 1 h after breakfast and its associated AUC were higher in the GDM compared with the NON-GDM group (effect of group, *p* = 0.01 and *p* = 0.02, respectively). At 2 h post-breakfast, the absolute glucose was similar between the GMD and NON-GDM groups (effect of group, *p* = 0.28), while the blood glucose AUC was higher in the GDM group (effect of group, *p* = 0.02). No absolute or AUC blood glucose values following breakfast were influenced by the LONG or SHORT exercise (effect of intervention, *p* > 0.05, Table 3).

#### 3.1.3. Post-Lunch Glucose Values

The absolute postprandial blood glucose within 1 and 2 h of lunch was higher in the GDM group compared with the NON-GDM group (effect of group, *p* = 0.03 and *p* = 0.04, respectively). Blood glucose AUC following lunch were similar between the GDM and NON-GDM groups up to 1 h (effect of group, *p* = 0.15) and 2 h postprandial (effect of group, *p* = 0.07). No absolute or AUC blood glucose values following lunch were influenced by the LONG or SHORT exercise (effect of intervention, *p* > 0.05, Table 3).

#### 3.1.4. Post-Dinner Glucose Values

When post-dinner blood glucose levels were assessed, the GDM group had higher absolute values at 1 h (effect of group, *p* = 0.03) but not 2 h post-dinner (effect of group, *p* = 0.38). The 1 and 2 h post-dinner AUC blood glucose values were higher for the GDM group compared with those for the NON-GDM group (effect of group, *p* = 0.04 and *p* = 0.04). Similar to breakfast and lunch, the post-dinner absolute and AUC blood glucose values were not altered by the SHORT or LONG exercise (effect of intervention, *p* > 0.05, Table 3).

### 3.2. Dietary Intake

There was a significant difference in the caloric and carbohydrate intake between groups whereby both were higher in the NON-GDM group compared with the GDM group (effect and group, *p* < 0.001 and *p* < 0.001, respectively). These group differences in caloric and carbohydrate intake were not influenced by the SHORT or LONG exercise (effect of intervention, *p* = 0.42 and *p* = 0.52, respectively). There were no differences between the GDM and NON-GDM groups for fat or protein intake (effect of group, *p* = 0.82 and *p* = 0.88, respectively), nor were these outcomes impacted by the exercise protocols (effect of intervention, *p* = 0.47 and *p* = 0.92, respectively, Table 4).

### 3.3. Physical Activity

The mean daily physical activity outcomes are presented in Table 5. The accelerometer wear time, total activity time, and time spent on each intensity of activity were similar between the GDM and NON-GDM groups (effect of group, *p* > 0.05). Furthermore, these outcomes were similar across all conditions, indicating that the SHORT and LONG protocols did not alter physical activity outcomes (effect of intervention, *p* > 0.05).

### 3.4. Exercise Compliance, Adherence, and Enjoyment

The GDM and NON-GDM groups were similarly compliant to the intervention regardless of the protocol, with both groups completing 76% and 90% of the SHORT walks and 87% and 93% of the LONG walks, respectively (effect of group, *p* = 0.31; effect of intervention, *p* = 0.49). The time spent walking per day was similar between the groups and across the SHORT and LONG conditions (effect of group, *p* = 0.98; effect of intervention, *p* = 0.91). The mean heart rates recorded during the prescribed walks were also similar between the GDM and NON-GDM groups for both SHORT and LONG protocols (effect of group, *p* = 0.17; effect of intervention, *p* = 0.43). Lastly, there were no differences noted between the groups or interventions for the PACES score, with the GDM and NON-GDM participants scoring similarly across the SHORT and LONG protocols (effect of group, *p* = 0.17; effect of intervention, *p* = 0.43, Table 6).

## 4. Discussion

Lifestyle interventions are considered front-line therapy for helping normalize blood glucose values in individuals diagnosed with GDM. However, the optimal timing of the exercise (pre- versus post-prandial) has not been established. We applied a randomized cross-over study design evaluating the effects of five days of postprandial walks (10 min immediately after breakfast, lunch, and dinner), and a single 30-min walk at any point following the first hour after a meal on 1 and 2 h postprandial and 24 h glucose outcomes in women with, and without GDM. Overall, there were no differences between the two exercise interventions regarding blood glucose control in pregnant individuals with or without GDM.

A systematic review and meta-analysis evaluating randomized controlled trials comparing exercise with no exercise in individuals with GDM (Davenport et al., 2018) found that chronic exercise was associated with reduced fasting blood glucose (standardized mean difference (SMD) −0.59, 95% CI −1.07 to −0.11; 4 RCTs, moderate effect size, 363 women) and postprandial blood glucose concentration (SMD −0.85, 95% CI −1.15 to −0.55; 3 RCTs, large effect size, 344 women) compared to the control group [32]. However, the timing and duration of the exercise for optimizing these results have garnered much less attention and had variable results. Diabetes Canada and the American College of Obstetricians and Gynecologists recommend that pregnant women diagnosed with GDM aim to complete 150 min of aerobic exercise per week, suggesting that multiple 10 min bouts could be as effective as a single longer session [2]. The ACOG recommends pregnant women with GDM take 10–15 min walks after eating; however, this recommendation was based on expert opinion and not empirical evidence [2]. To date, only three studies have examined the acute effects of exercise on glucose metabolism in women with GDM [31,32,33]. Lesser et al. (1996) prescribed a single bout of exercise (30 min cycling at 60% VO_2_max) followed by a mixed-nutrient breakfast 14 h later in individuals with and without GDM [33]. The authors observed no impact of the exercise on postprandial glycemic control, fasting or peak glucose values, insulin levels, or the area under the curve of glucose. Avery and Walker (2001) prescribed a mixed-nutrient meal to individuals with GDM, followed by a 90 min break [31]. The participants were then asked to either rest for 30 min or complete 30 min of low- or moderate-intensity exercise, followed by an additional 90 min of rest. Both exercise conditions resulted in a reduction in blood glucose 15 min following the cessation of exercise (but not at any other time point) compared to rest. Finally, Garcia-Patterson et al. (2001) prescribed self-paced walking for an hour after a standard breakfast for individuals with GDM [34]. Compared to the control condition of no walking, 1 h postprandial glucose was reduced and was thus positively impacted by the exercise [33]. These data imply that post- and not pre-prandial aerobic exercise influences glycemic control. That said, our study failed to show any influence of exercise on blood glucose in GDM, and as such, the data are largely inconclusive. From a clinical perspective, while Avery and Walker (2001) and Garcia-Patterson et al. (2001) showed an effect of the exercise on blood glucose, the feasibility of the protocols used would likely be challenging for pregnant individuals to adopt, as time and resources are often common barriers to exercise participation [31,34].

The first study to compare three 10 min postprandial walks to 30 min continuous moderate intensity walking at any time in individuals with GDM found that the walking had no impact on 3 h postprandial glucose values [25]. The lack of effectiveness of the SHORT or LONG exercise on postprandial glucose in either study may reflect the overall short duration of the intervention (3 days versus 5 days). Indeed, most exercise studies that show a reduction in fasting and 24 h blood glucose levels in pregnant women with or without GDM typically takes place over a much longer period—generally, about 6 weeks minimum [19,20,35]). Nonetheless, chronic engagement in intermittent or continuous exercise has yet to be established for individuals with GDM. In doing so, a more fulsome understanding of the impact of differing exercise volumes (intensity and/or duration) and their impact on glucose outcomes (peak glucose, time > 7.8 mmol/L, postprandial values, 24 h mean) can be generated.

Regarding dietary intake, as a front-line therapy for managing GDM, dietary intake is important in controlling blood glucose values. As such, GDM patients often receive dietary counselling and are advised to consume foods with lower glycemic indices and control their caloric intake [36,37]. It is therefore unsurprising that, across all conditions, women in the GDM group consumed significantly lower amounts of carbohydrates and calories compared to their counterparts. The conscious effort to reduce the caloric and carbohydrate intake in the GDM group may have already helped blunt their glycemic response to a meal and thus may hide the potential effect of the postprandial timing of exercise.

The high level of compliance to the intervention was reflected in the results of the physical activity enjoyment scale, which indicate that, within each group, neither the three 10 min walks nor the 30 min walk were more enjoyable. Given that walking is a feasible, accessible, and low-cost activity and that physical activity is important in glycemic control, enjoyment can be equally attained regardless of the approach in terms of duration or timing around a meal [26,32]. As alluded to by Christie and colleagues (2022), it is encouraging that 10 min walks could be prescribed for individuals with GDM, since it yielded similar glucose outcomes compared to 20 min continuous walking [25]. This is particularly beneficial for those facing barriers to exercise including fatigue, discomfort, and time [38]. This may help pregnant individuals maintain their physical activity behaviors throughout pregnancy and provide a rationale for exercise practitioners to encourage at least short exercise bouts to those with gestational diabetes. As the research continues to evolve with respect to exercise and gestational diabetes, it is anticipated that the specificity of exercise prescription including the duration, frequency, and intensity can be further improved.

### 4.1. Strengths

We recruited a heterogenous sample of singleton pregnant individuals that were all of a similar gestational age. It is noteworthy that this is the first study to investigate whether the timing of exercise has any implication on blood glucose control in GDM compared with NON-GDM pregnant individuals. Furthermore, we used heart rate monitoring to confirm the exercise intensity, which is an important consideration particularly when prescribing exercise expectations. This study is of high external validity since participants were in free-living conditions and thus helps to make the results more clinically meaningful. Lastly, the prescribed exercise conditions are practical because walking is typically an accessible activity, using minimal equipment, and a total of 30 min per day is achievable in terms of time commitment compared to the 60 min walking condition investigated by [34].

### 4.2. Limitations

First, the method of monitoring glucose control analysis may be limited in accuracy since it reports on interstitial blood glucose and may underestimate values if compared to direct finger poke blood samples. In line with this, we also cannot comment on pre-post exercise values since the CGM records data every 15 min. Additionally, daily 24 h glucose data are excluded from the first day of each condition to avoid including pre-stimulus values; however, this may bias the results towards a more effective intervention due to the chronic benefits and dose–response relationship of exercise. The study is also limited by evaluating only moderate-intensity exercise regarding postprandial glucose, and as such, our conclusions are exclusive to walking at this intensity only. Nonetheless, we prescribed as per current prenatal exercise guidelines to help manage blood glucose values. Future studies are necessary to explore other exercise intensities and, potentially, durations regarding postprandial glucose. Research is also needed to determine whether exercise timing could help prevent GDM, which would be clinically important for all pregnancies. Lastly, although the caloric and carbohydrate intake was similar across all conditions within each group, not implementing standardized meals could mean that some postprandial values were affected by dietary compositions.

## 5. Conclusions

In conclusion, moderate intensity walking either three times daily for 10 min immediately after eating or for 30 min daily at least 1 h after food intake does not influence blood glucose control in GDM and NON-GDM individuals.

## Figures and Tables

**Figure 1 ijerph-20-05500-f001:**
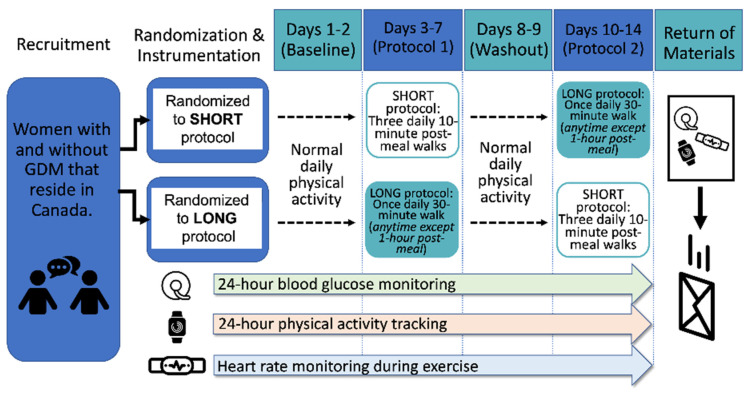
Study design schematic.

**Table 1 ijerph-20-05500-t001:** Participant Characteristics.

Participant Characteristics	GDM Group (*N* = 17)	NON-GDM Group (*N* = 16)	*p*
Age (years)	33 ± 5	32 ± 2	0.37
Gestational age at participation (weeks)	33 ± 2	32 ± 3	0.12
Height (cm)	163 ± 12	163 ± 7	0.99
Body mass at participation (kg)	87 ± 18	78 ± 18	0.18
BMI at participation (kg/m^2^)	33 ± 7	29 ± 6	0.11
Pre-pregnancy body mass (kg)	77 ± 18	69 ± 18	0.26
Pre-pregnancy BMI (kg/m^2^)	29 ± 7	26 ± 6	0.17
Parity	1 ± 1	0 ± 1	0.18
Previous history of GDM (*N* (%))	4 (25)	0	-
** *Ethnicity* **			
Asian	1	4	
South Asian	0	1	
Black/African American	3	2	
Caucasian	8	7	
Hispanic	1	1	
Mauritian	1	0	
Mixed heritage	3	1	

Data presented as mean ± SD unless otherwise indicated. GDM; gestational diabetes mellitus; NON-GDM, individuals without gestational diabetes mellitus; BMI, body mass index; GDM, gestational diabetes mellitus.

**Table 2 ijerph-20-05500-t002:** Fasting, 24 h, peak, and nadir values and time in target, time below 3.3 mmol/L, and time above 7.8 mmol/L in women with and without GDM across three conditions.

	GDM Group (*N* = 14)			NON-GDM Group (*N* = 15)			*p*-Values		
	*Normal*	*Short*	*Long*	*Normal*	*Short*	*Long*	*Group*	*Intervention*	*Interaction*
**Fasting** (mmol/L)	3.95 ± 0.57 *	3.97 ± 0.53 *	3.85 ± 0.58 *	3.58 ± 0.36	3.64 ± 0.43	3.79 ± 0.36	**0.02**	0.90	0.41
**24 h mean** (mmol/L)	4.54 ± 0.52 *	4.72 ± 0.52 *	4.43 ± 0.46 *	4.27 ± 0.35	4.30 ± 0.44	4.45 ± 0.38	**0.02**	0.69	0.16
**Peak** (mmol/L)	7.0 ± 1.0 *	7.1 ± 1.0 *	6.9 ± 0.7 *	6.53 ± 1.1	6.44 ± 0.90	6.67 ± 0.86	**0.03**	0.99	0.69
**Nadir** (mmol/L)	3.2 ± 0.52	3.4 ± 0.48	3.1 ± 0.58	3.0 ± 0.23	3.0 ± 0.45	3.2 ± 0.30	0.11	0.48	0.07
**Time in target** (min)	1307 ± 141	1311 ± 152	1267 ± 213	1303 ± 120	1281 ± 172	1352 ± 98	0.60	0.95	0.33
**Time < 3.3** (min)	114 ± 149	76 ± 141	137 ± 185	130 ± 119	147 ± 178	48 ± 84	0.98	0.73	0.11
**Time > 7.8** (min)	13 ± 26	15 ± 21	8 ± 15	7 ± 16	5 ± 8	9 ± 16	0.17	0.94	0.50

Data presented as the daily mean ± SD. GDM, gestational diabetes mellitus; NON-GDM, individuals without gestational diabetes mellitus. * = statistically significant difference from the NON-GDM group, *p* < 0.05.

**Table 3 ijerph-20-05500-t003:** 1 h and 2 h postprandial glucose values in women with and without GDM across three conditions.

Postprandial Glucose	GDM Group (*N* = 14)			NON-GDM Group (*N* = 15)			*p*-Values		
	*Normal*	*Short*	*Long*	*Normal*	*Short*	*Long*	*Group*	*Intervention*	*Interaction*
**1 h post-breakfast** (mmol/L)	5.41 ± 1.0	5.63 ± 1.1	5.25 ± 0.7	4.8 ± 0.9	4.9 ± 0.9	5.1 ± 0.6	** *0.01* **	0.79	0.50
**1 h post-breakfast AUC** (mg/dL)	316 ± 47	344 ± 69	311 ± 40	297 ± 49	299 ± 48	299 ± 39	** *0.02* **	0.38	0.40
**2 h post-breakfast** (mmol/L)	4.6 ± 0.6	4.6 ± 0.4	4.4 ± 0.4	4.2 ± 0.6	4.4 ± 0.5	4.6 ± 0.8	0.28	0.78	0.20
**2 h post-breakfast AUC** (mg/dL)	612 ± 89	648 ± 110	597 ± 67	568 ± 83	574 ± 81	584 ± 68	** *0.02* **	0.56	0.40
**1 h post-lunch** (mmol/L)	5.4 ± 1.1	5.3 ± 0.6	5.4 ± 0.9	4.9 ± 0.8	4.9 ± 0.8	5.2 ± 0.7	** *0.03* **	0.63	0.80
**1 h post-lunch AUC** (mg/dL)	307 ± 53	313 ± 40	306 ± 46	292 ± 45	290 ± 42	304 ± 37	0.15	0.89	0.67
**2 h post-lunch** (mmol/L)	4.9 ± 0.8	5.1 ± 0.6	5.0 ± 0.7	4.8 ± 0.6	4.6 ± 0.4	4.9 ± 0.5	** *0.04* **	0.73	0.37
**2 h post-lunch AUC** (mg/dL)	615 ± 104	624 ± 71	621 ± 88	582 ± 82	575 ± 67	606 ± 61	0.07	0.73	0.72
**1 h post-dinner** (mmol/L)	5.5 ± 0.8	5.2 ± 0.7	5.3 ± 0.9	4.9 ± 0.7	5.0 ± 0.8	5.0 ± 0.8	** *0.03* **	0.83	0.71
**1 h post-dinner AUC** (mg/dL)	316 ± 37	322 ± 48	310 ± 52	292 ± 43	295 ± 41	302 ± 39	** *0.04* **	0.92	0.65
**2 h post-dinner** (mmol/L)	5.2 ± 0.9	4.8 ± 0.9	4.8 ± 0.8	4.9 ± 0.7	4.7 ± 0.9	4.8 ± 0.9	0.38	0.46	0.77
**2 h post-dinner AUC** (mg/dL)	633 ± 77	636 ± 83	614 ± 92	586 ± 82	583 ± 76	599 ± 86	** *0.04* **	0.98	0.64

Data presented as the daily mean ± SD. GDM; gestational diabetes mellitus; NON-GDM, individuals without gestational diabetes mellitus; AUC, area under curve.

**Table 4 ijerph-20-05500-t004:** Mean daily dietary intake in women with and without GDM throughout the NORMAL, SHORT, and LONG conditions.

	GDM Group (*N* = 14)			NON-GDM (*N* = 15)			*p*-Values		
	*Normal*	*Short*	*Long*	*Normal*	*Short*	*Long*	*Group*	*Intervention*	*Interaction*
**kcals**	2026 ± 355	2031 ± 336	2161 ± 452	2375 ± 500	2684 ± 608	2368 ± 487	**<0.001**	0.42	0.17
**Carbs** (g)	188 ± 52	197 ± 57	206 ± 67	299 ± 68	326 ± 67	288 ± 71	**<0.001**	0.52	0.38
**Fat** (g)	104 ± 29	99 ± 29	109 ± 26	93 ± 23	116 ± 37	99 ± 24	0.82	0.47	0.12
**Protein** (g)	98 ± 24	93 ± 21	97 ± 25	94 ± 21	100 ± 19	92 ± 20	0.88	0.92	0.45

Data presented as the daily mean ± SD. GDM, gestational diabetes mellitus; NON-GDM, individuals without gestational diabetes mellitus; Kcals, kilocalories.

**Table 5 ijerph-20-05500-t005:** Physical activity outcomes during prescribed walking time in women with and without GDM throughout the SHORT and LONG conditions.

	GDM Group (*N* = 14)			NON-GDM Group (*N* = 15)			*p*-Values		
	Normal	Short	Long	Normal	Short	Long	*Group*	*Intervention*	*Interaction*
**Wear time (min/d)**	846 ± 71	834 ± 81	800 ± 66	835 ± 95	884 ± 131	881 ± 113	0.06	0.71	0.21
**Sedentary (min)**	585 ± 113	544 ± 93	507 ± 87	535 ± 106	573 ± 119	591 ± 105	0.37	0.91	0.07
**Sedentary (%)**	69 ± 11	65 ± 10	64 ± 11	64 ± 10	65 ± 11	67 ± 9	0.79	0.88	0.30
**Light (min)**	233 ± 67	255 ± 60	250 ± 61	260 ± 65	254 ± 67	239 ± 73	0.72	0.83	0.56
**Light (%)**	28 ± 8	31 ± 6	31 ± 6	31 ± 8	29 ± 8	27 ± 8	0.78	0.95	0.16
**Moderate (min)**	28 ± 44	34 ± 39	40 ± 49	22 ± 12	30 ± 15	35 ± 15	0.50	0.38	0.99
**Moderate (%)**	3 ± 5	4 ± 4	5 ± 5	3 ± 2	4 ± 2	4 ± 2	0.51	0.31	0.99
**Vigorous (min)**	0.3 ± 1.0	0.7 ± 0.8	2.5 ± 6.1	0.2 ± 0.4	0.3 ± 0.5	0.3 ± 0.3	0.15	0.26	0.28
**Vigorous (%)**	0.0 ± 0.1	0.1 ± 0.1	0.3 ± 0.1	0.0 ± 0.1	0.0 ± 0.1	0.0 ± 0.0	0.15	0.26	0.28
**Total active time (min)**	261 ± 106	289 ± 96	290 ± 102	289 ± 69	285 ± 63	238 ± 116	0.65	0.66	0.29
**Total active time (%)**	31 ± 11	35 ± 10	36 ± 10	35 ± 9	33 ± 8	31 ± 8	0.73	0.94	0.24

Data presented as the daily mean ± SD. GDM, gestational diabetes mellitus; NON-GDM, individuals without gestational diabetes mellitus.

**Table 6 ijerph-20-05500-t006:** Compliance and adherence to the exercise intervention for LONG and SHORT protocols between GDM and NON-GDM.

	GDM Group (*N* = 14)		NON-GDM Group (*N* = 15)			*p*-Values	
	Short	Long	Short	Long	*Group*	*Intervention*	*Interaction*
**Walks completed (%)**	76 ± 37	87 ± 33	90 ± 16	93 ± 21	0.31	0.49	0.66
**Walk time per day (min)**	30 ± 7	32 ± 5	32 ± 6	30 ± 2	0.98	0.91	0.20
**Mean heart rate (bpm)**	116 ± 11	117 ± 12	120 ± 11	120 ± 8	0.41	0.84	0.94
**PACES Score**	91 ± 23	97 ± 18	100 ± 14	102 ± 16	0.17	0.43	0.69

Data presented as the daily mean ± SD. GDM, gestational diabetes mellitus; NON-GDM, individuals without gestational diabetes mellitus; min, minutes; BPM, beats per minute; PACES, Physical activity enjoyment scale.

## Data Availability

Data are contained within the article.

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
