# Peer review of "Optimizing Blood Glucose Control through the Timing of Exercise in Pregnant Individuals Diagnosed with Gestational Diabetes Mellitus"

_ijerph, 2023, doi:10.3390/ijerph20085500_

Round 1

Reviewer 1 Report

This is an interesting study for clinical exercise physiology practice that examines the post-prandial influence of a continuous compared to intermittent exercise format in patients of gestational glucose intolerance. The manuscript is well written and does not require substantial modifications in that regard . Although inclusion of a series of graphs would make the presentation of results much clearer as the tables are quite extensive and at times do not focus on the important pieces of data. The methods are comprehensive. The introduction should be modified to include a paragraph focusing on the literature concerning postprandial responses to diet and exercise interventions . The discussion section should clearly focus on the clinical implications for exercise practitioners when working with patients with gestational diabetes. In addition the discussion section should also include implications for chronic adaptations where intermittent (‘broken’) or continuous exercise is administered. Clearly the limitation of the study it that exercise intensity and s not explored on post-prandial responses. That should also be a focus of the discussion section.

Author Response

Reviewer 1 comments

Comment #1:

This is an interesting study for clinical exercise physiology practice that examines the post-prandial influence of a continuous compared to intermittent exercise format in patients of gestational glucose intolerance. The manuscript is well written and does not require substantial modifications in that regard.

Response #1:

Thank you for taking the time to review our manuscript. We are pleased that it has been well received.

Comment #2:

Although inclusion of a series of graphs would make the presentation of results much clearer as the tables are quite extensive and at times do not focus on the important pieces of data. The methods are comprehensive.

Response #2:

Thank you for your suggestion. We deliberated about this prior to initial submission and concluded that given the groupings (2 groups with 6 time points), a series of graphs in this instance might be more confusing. As per Reviewer 2s suggestion, we have provided clarity within the tables to improve their readability.

Comment #3:

The introduction should be modified to include a paragraph focusing on the literature concerning postprandial responses to diet and exercise interventions.

Response #2:

Thank you for this suggestion; we have added a short paragraph to address this comment.

Lifestyle interventions that aim to improve blood glucose control in GDM have largely comprised of changes to diet with and without exercise. Allendhan et al. (2019) collated data from 8 trials that compared the effect of diet plus exercise, against diet only interventions on postprandial blood glucose (Avery et al., 1997; Bo et al., 2014; Brankston et al., 2004; De Barros et al., 2010; Halse et al., 2014a; Jovanovic-Peterson et al., 1989; Kokic et al., 2018; Youngwanichsetha et al., 2014). Of the trials, six evidenced lower postprandial blood glucose control when diet and exercise interventions (3 aerobic, 1 yoga, 1 resistance and 1 aerobic and resistance exercise combined), were combined versus dietary changes only (Bo et al., 2014; Brankston et al., 2004; Halse et al., 2014a; Jovanovic-Peterson et al., 1989; Kokic et al., 2018; Youngwanichsetha et al., 2014). This implies that the exercise stimulus is a primary driver, via non-insulin mediated mechanisms, in helping to manage postprandial blood glucose in GDM. It is understood exercise increases the rate of glucose uptake into the skeletal muscle during and following exercise. This increased uptake occurs due to the translocation of the glucose transport protein GLUT-4 from intracellular sites to the sarcolemma and T-tubules. This action increases the sites at which glucose can diffuse into the muscle cell and thus, reduces the level of glucose in the blood (Barnard & Youngren, 1992; Bird et al., 2017). To date, the isolated effect of exercise on blood glucose control in GDM is poorly characterised. (Page 4)

Comment #4: The discussion section should clearly focus on the clinical implications for exercise practitioners when working with patients with gestational diabetes.

Response #4: Thank you for your suggestion, we have now added some information to the discussion.

From a clinical perspective, while Avery and Walker (2001) and Garcia-Patterson et al. (2001) showed an effect of the exercise on blood glucose, the feasibility of the protocols used would likely be challenging for pregnant individuals to adopt where time and resources are often common barriers to exercise participation. (Page 18)

This may help pregnant individuals maintain their physical activity behaviours throughout pregnancy and provides rationale for exercise practitioners to encourage at least short exercise bouts to those with gestational diabetes. As the research continues to evolve with respect to exercise and gestational diabetes, it is anticipated that the specificity of exercise prescription including duration, frequency and intensity can be further improved. (Page 18)

Comment #5: In addition, the discussion section should also include implications for chronic adaptations where intermittent (‘broken’) or continuous exercise is administered.

Response #5: We apologize in advance; however, it is not totally clear what is being asked of the authors here. The intervention we administered did involve intermittent exercise (three 10-minute walks), and continuous exercise (a 30-minute walk per day). In the discussion we acknowledge that longer studies do see greater changes. Longer study durations are necessary to validate the prenatal exercise guidelines that are in place. 

The lack of effectiveness of the SHORT or LONG exercise on postprandial glucose in either study may reflect the overall short duration of the intervention (3 days versus 5 days). Indeed, most exercise studies that show a reduction in fasting and 24h blood glucose levels in pregnant women with or without GDM, typically take place over a much longer period, generally about 6 weeks minimum (Cordero et al., 2015a; Halse et al., 2014b; Jovanovic-Peterson et al., 1989).  Nonetheless, chronic engagement in intermittent or continuous exercise has yet to be established for individuals with GDM. In doing so, a more fulsome understanding of the impact of differing exercise volumes (intensity and/or duration), and its impact on glucose outcomes (peak glucose, time > 7.8 mmol/L, postprandial values, 24h mean) can be generated. (Page 18 – 19)

Comment #6: Clearly the limitation of the study it that exercise intensity and is not explored on post-prandial responses. That should also be a focus of the discussion section.

Response #6: The authors infer from your comment that the limitation of the study is that exercise intensity is not explored on post-prandial responses. The purpose of our study was to explore moderate intensity exercise since this is what the guidelines recommend. Further research exploring different intensities would certainly elucidate the impact of exercise on post-prandial glucose further.

Reviewer 2 Report

Thank you for the opportunity to review your manuscript, Optimizing blood glucose control through timing of exercise in pregnant individuals diagnosed with gestational diabetes mellitus.

First of all, congratulations to the authors, as the article is well written, easy to follow in its writing and methodology, and the presentation of the results.

Minor issues

The last sentence of the conclusion should be moved to the discussion section, as future studies or lines of research to follow, as it cannot be considered a conclusion is drawn from the study.

Other minor issues are some red words and a blank page at the end that needs to be revised.

Some authors need their affiliation.

Author Response

Reviewer 2 comments

Comment #1: Thank you for the opportunity to review your manuscript, Optimizing blood glucose control through timing of exercise in pregnant individuals diagnosed with gestational diabetes mellitus. First of all, congratulations to the authors, as the article is well written, easy to follow in its writing and methodology, and the presentation of the results.

Response #1: Thank you for taking the time to review our manuscript. We are pleased that it has been well received and understood.

Comment #2: Minor issues - The last sentence of the conclusion should be moved to the discussion section, as future studies or lines of research to follow, as it cannot be considered a conclusion is drawn from the study.

Response #2: The last sentence has now been moved to the last paragraph of the discussion.

The study is also limited by evaluating only moderate intensity exercise on postprandial glucose and as such, our conclusions are exclusive to walking at this intensity only. Nonetheless, we prescribed as per current prenatal exercise guidelines to help manage blood glucose values. Future studies are necessary explore other exercise intensities and potentially, durations, on postprandial glucose. Research is also needed to determine whether exercise timing could help prevent GDM, which would be clinically important for all pregnancies. (Page 19)

Comment #3: Other minor issues are some red words and a blank page at the end that needs to be revised. Some authors need their affiliation.

Response #3: Thank you for highlighting this. These issues have now been resolved.

Reviewer 3 Report

This paper provides findings from a randomized, cross-over study to determine the impact of two postprandial physical activity prescriptions blood glucose levels among GDM and non-GDM women. The study is important as non-medication blood sugar control strategies improves patient and infant birth outcomes. 

1. I am not sure I understand from the introduction the rationale for the current study. To me it it seems like you are comparing two prescriptions for PA for blood sugar management. Not clear how the non-GDM participants fit into the hypothesis testing. Furthermore, background regarding the two prescriptions as justification for the current study would improve rationale.

2. The second paragraph in the '2.2 Participants' section is confusing.

3. I wondering about inclusion and exclusion criteria, given the means from which some patients were recruited (i.e. social media)

4. What is the rationale for the study design protocol. Is 4 days sufficient enough to observe study effects?

5. I don't understand the cross-over design rationale. Both groups were receiving an intervention.

6. Table 2 does not give me a clear idea of the starting values and changes post intervention. Also, I don't understand who was in the NORMAL VS. SHORT VS LONG groups, or how many participants are in each. I would be concerned with power given the starting number is so low. Also what is time in target, time <3.3 and time >7.8 how does that relate to the research question of effectiveness.

7. I feel like the findings could be better presented in a way that is tied to the research questions being asked. In general, when you are making this type of comparison you are looking at pre and post intervention values to determine who has responded to the intervention.

8. It seems like the authors are presenting a lot of additional findings that obfuscate the major findings as they relate to the question of whether two PA approaches lead to significant changes in blood sugar control. 

Author Response

Reviewer 3 comments

Comment #1: This paper provides findings from a randomized, cross-over study to determine the impact of two postprandial physical activity prescriptions blood glucose levels among GDM and non-GDM women. The study is important as non-medication blood sugar control strategies improves patient and infant birth outcomes. 

Response #1: Thank you for taking the time to review our manuscript.

Comment #2: I am not sure I understand from the introduction the rationale for the current study. To me it seems like you are comparing two prescriptions for PA for blood sugar management. Not clear how the non-GDM participants fit into the hypothesis testing. Furthermore, background regarding the two prescriptions as justification for the current study would improve rationale.

Response #2: Thank you for your suggestion. We have added some text to the third paragraph of the introduction. The intention with this addition was to reiterate the relevance of the study to gestational diabetes,

Recently, the American College of Obstetrics and Gynaecology (ACOG) suggested that a 10-15 minute walk after each meal may improve glycaemic control; an outcome that is of particular importance for individuals with GDM (Bulletins-Obstetrics, 2018). (Page 3)

We have also added text to improve the clarity of information surrounding the two exercise prescriptions formats.

However, this exercise recommendation is largely based on expert opinion and not empirical evidence. As such, there is a need to distinguish between the effectiveness of intermittent, or accumulated exercise bouts that equates to 150-minutes per week on glycaemic control for individuals with GDM. (Page 3)

Lastly, we do have a sentence in to justify the inclusion of the non-GDM group which we perceive to be sufficient.

To build on this knowledge, we herein perform a similar protocol with the addition of a normoglycemic control group to elucidate any group differences that may emerge. (Page 4)

Comment #2: The second paragraph in the '2.2 Participants' section is confusing.

Response #2: We have reworded paragraph 2.2 and have moved the sample size information to the statistical analysis section as it may be more appropriately placed there. This paragraph now reads:

To be included in the study, participants were required to be residents of Canada, pregnant with one baby, and had either a diagnosis of GDM, or an uncomplicated pregnancy. Participants were recruited after 20 weeks’ gestation since GDM is generally screened for and diagnosed toward the end of the second trimester of pregnancy. Individuals were excluded if they had absolute contraindications to prenatal exercise as identified by the PARmed-X for Pregnancy. These contraindications included premature labour, placenta previa, pregnancy-induced hypertension, pre-eclampsia, high—order pregnancy uncontrolled systemic disease including cardiovascular and respiratory disorders (Wolfe and Mottola, 2002). Participants were recruited by a physician working in a GDM clinic and, via social media advertisements on Facebook and Instagram. Posters and pre-recorded video presentations were also created and distributed to relevant provincial, and national health and education clinics specialising in diabetes and pregnancy. (Page 5)

Comment #3:  I wondering about inclusion and exclusion criteria, given the means from which some patients were recruited (i.e. social media)

We anticipate that this (inclusion and exclusion criteria) has been made clearer by the changes made to the paragraph on participants. To reiterate, we have added more specific information regarding the exclusion criteria.

Individuals were excluded if they had absolute contraindications to prenatal exercise as identified by the PARmed-X for Pregnancy. These contraindications included premature labour, placenta previa, pregnancy-induced hypertension, pre-eclampsia, high—order pregnancy uncontrolled systemic disease including cardiovascular and respiratory disorders (Wolfe and Mottola, 2002). (Page 5)

Comment #4: What is the rationale for the study design protocol? Is 4 days sufficient enough to observe study effects?

Response #4: In total, participants engaged in 10 days of walking; 5 days with 3 10-minute walks and 5 days with 30 minutes continuous walking with both protocols separated by 2 days of normal activity levels only. To date, it is not clear if the study period is the limitation to observing any post-prandial effects. However, our study is longer than most that attempted to understand the acute effects of exercise on post-prandial glucose and these are discussed within the second paragraph of the discussion. In the third paragraph we also acknowledge that the study duration may limit our ability to capture any effect, however Christie et al. (2022) used the longest intervention to date (a 3-day protocol), so we still have a longer intervention.

The first study to compare three 10-minute postprandial walks to 30-minutes continuous moderate intensity walking at any time in individuals with GDM found that the walking had no impact on 3-hour postprandial glucose values (Christie et al., 2022). The lack of effectiveness of the SHORT or LONG exercise on postprandial glucose in either study may reflect the overall short duration of the intervention (3 days versus 5 days). (Page 18)

Nonetheless, we draw upon previous studies that adopt a more chronic approach.

The lack of effectiveness of the SHORT or LONG exercise on postprandial glucose in either study may reflect the overall short duration of the intervention (3 days versus 5 days). Indeed, most exercise studies that show a reduction in fasting and 24h blood glucose levels in pregnant women with or without GDM, typically take place over a much longer period, generally about 6 weeks minimum (Cordero et al., 2015; Halse et al., 2014b; Jovanovic-Peterson et al., 1989).  Nonetheless, chronic engagement in intermittent or continuous exercise has yet to be established for individuals with GDM. In doing so, a more fulsome understanding of the impact of differing exercise volumes (intensity and/or duration), and its impact on glucose outcomes (peak glucose, time > 7.8 mmol/L, postprandial values, 24h mean) can be generated. (Page 18)

Comment #5: I don't understand the cross-over design rationale. Both groups were receiving an intervention.

Response #5: The cross-over design was to ensure that interpretation of results was not limited by the order of protocol (SHORT or LONG). It is also best practice to implement a cross-over design where both groups participated in two intervention approaches. A sentence has been added to the section about the Exercise Protocol to improve transparency here.

Following completion of two NORMAL days to establish baseline blood glucose and activity monitoring, all participants completed the SHORT and LONG protocol in a randomised order. This randomised approach was taken to ensure that the interpretation of results were not limited by the sequence of protocol. Women randomized to the SHORT condition were asked to complete a 10-minute walk within in the first hour after breakfast, lunch and dinner totalling three 10-minute walks per day for five days. Those randomized to the LONG condition were asked to complete 30 minutes of walking at any time of day, other than the hour immediately following breakfast, lunch or dinner for five days. Between interventions, participants completed two more days of NORMAL activity that fulfilled a washout period between protocols. Following the two-day washout period, participants were asked to complete the complementary exercise protocol for five days. (Page 7)

Comment #6: Table 2 does not give me a clear idea of the starting values and changes post intervention. Also, I don't understand who was in the NORMAL VS. SHORT VS LONG groups, or how many participants are in each. I would be concerned with power given the starting number is so low. Also what is time in target, time <3.3 and time >7.8 how does that relate to the research question of effectiveness.

Response #6: Thank you for your comment. Firstly, we anticipate that the addition of the sentence on Page 7 will add clarity to your comment.

All participants completed the SHORT and LONG protocol however the order of the interventions were randomised. This approach was taken to ensure that the interpretation of results were not limited by the sequence of protocol. (Page 7)

For improved clarity, we have added information to the Study Structure and Exercise Protocol Section.

The study period began with two days of normal daily physical activity [NORMAL], followed by five consecutive days of the first intervention condition, then a washout period of again, two days of NORMAL daily physical activity, followed by five days of the second intervention condition (see Figure 1). The interventions comprised of three 10-minute walks per day for five days [SHORT], or one 30-minute walk each day, for 5 days [LONG]. (Page 6 and 7)

Following completion of two NORMAL days to establish baseline blood glucose and activity monitoring, all participants completed the SHORT and LONG protocol in a randomised order. This randomised approach was taken to ensure that the interpretation of results were not limited by the sequence of protocol. Women randomized to the SHORT condition were asked to complete a 10-minute walk within in the first hour after breakfast, lunch and dinner totalling three 10-minute walks per day for five days. Those randomized to the LONG condition were asked to complete 30 minutes of walking at any time of day, other than the hour immediately following breakfast, lunch or dinner for five days. Between interventions, participants completed two more days of NORMAL activity that fulfilled a washout period between protocols. Following the two-day washout period, participants were asked to complete the complementary exercise protocol for five days.

To improve the understanding of Table 2, we have more clearly separated the GDM and NON-GDM groupings.

While the numbers may be low, our sample is similar to that previously used by Christie et al. (2022) using an almost identical study design. We also have a paragraph explaining our sample size and therefore, feel justified in the numbers that we have. To reiterate:

Avery and Walker (2001) found that low-intensity postprandial exercise resulted in a mean difference (±SD) in blood glucose of 0.3 ± 0.3 at 30 minutes post- exercise (Avery & Walker, 2001a). Based on these findings, we estimated that 12 women are required per group to observe a significant difference in postprandial blood glucose and increased the required sample size by 20% to account for study withdrawal (n=15 per group; 80% power, α = 0.05; G*Power v3.1.9). (Page 12)

Lastly, time in target (3.3-7.8 mmol/L), time spent < 3.3 mmol/L, and time spent > 7.8 mmol/L are as described and are secondary outcomes. This data is generated by the CGM and we perceive them to be important variables to monitor, particularly to ensure risk of hypoglycaemia is not impacted by the intervention.

Comment #7: I feel like the findings could be better presented in a way that is tied to the research questions being asked. In general, when you are making this type of comparison you are looking at pre and post intervention values to determine who has responded to the intervention.

Response #7: Thank you for your comment. We agree that pre-post intervention values are atypical with this type of study design. However, the glucose monitor is limited to just 14 days of data and we prioritised a baseline and washout period over the post-intervention measurement. That said, since it was randomised, post intervention data will have been accounted for in the NORMAL days which in turn solidifies the rationale for randomising the protocols. Additionally, the way our data is presented is similar to Christie et al. (2022) and therefore is useful when comparing the two studies.

Comment #8: It seems like the authors are presenting a lot of additional findings that obfuscate the major findings as they relate to the question of whether two PA approaches lead to significant changes in blood sugar control.

Response #8: We agree there is a lot of data presented in the paper. In the methods section we do describe both the primary and secondary outcomes.

The primary outcomes were 1- and 2-hour postprandial glucose values after the start of each meal (breakfast, lunch, dinner). Secondary outcomes included fasting (value upon awakening), mean 24-hour (midnight to midnight), peak and nadir glucose, time in target (3.3-7.8 mmol/L), time spent < 3.3 mmol/L, and time spent > 7.8 mmol/L. (Page 9)

We believe that given the infancy of our understanding regarding blood glucose control in response to exercise timing, data transparency is important given the level of detail that the CGM can provide.